# Increased power from conditional bacterial genome-wide association identifies macrolide resistance mutations in *Neisseria gonorrhoeae*

Kevin C. Ma [1], Tatum D. Mortimer [1], Marissa A. Duckett[1], Allison L. Hicks [1], Nicole E. Wheeler [2], Leonor Sánchez-Busó [2] & Yonatan H. Grad [1,3]✉

The emergence of resistance to azithromycin complicates treatment of *Neisseria gonorrhoeae*, the etiologic agent of gonorrhea. Substantial azithromycin resistance remains unexplained after accounting for known resistance mutations. Bacterial genome-wide association studies (GWAS) can identify novel resistance genes but must control for genetic confounders while maintaining power. Here, we show that compared to single-locus GWAS, conducting GWAS conditioned on known resistance mutations reduces the number of false positives and identifies a G70D mutation in the RplD 50S ribosomal protein L4 as significantly associated with increased azithromycin resistance ($p$-value = $1.08 \times 10^{-11}$). We experimentally confirm our GWAS results and demonstrate that RplD G70D and other macrolide binding site mutations are prevalent (present in 5.42% of 4850 isolates) and widespread (identified in 21/65 countries across two decades). Overall, our findings demonstrate the utility of conditional associations for improving the performance of microbial GWAS and advance our understanding of the genetic basis of macrolide resistance.

[1] Department of Immunology and Infectious Diseases, Harvard T.H. Chan School of Public Health, Boston, MA, USA. [2] Centre for Genomic Pathogen Surveillance, Wellcome Sanger Institute, Wellcome Genome Campus, Hinxton, Cambridgeshire, UK. [3] Division of Infectious Diseases, Brigham and Women's Hospital and Harvard Medical School, Boston, MA, USA. ✉email: ygrad@hsph.harvard.edu

Increasing antibiotic resistance in *Neisseria gonorrhoeae*, the causative agent of the sexually transmitted disease gonorrhea, threatens effective control of this prevalent pathogen[1–3]. Current empiric antibiotic therapy in the US comprises a combination of the cephalosporin ceftriaxone and the macrolide azithromycin, but increasing prevalence of azithromycin resistance has led some countries, such as the UK, to instead recommend ceftriaxone monotherapy[4]. Rapid genotypic diagnostics for antimicrobial susceptibility have been proposed as a platform to tailor therapy and to extend the clinically useful lifespan of anti-gonococcal antibiotics[5,6]. These rapid diagnostics rest on robust genotype-to-phenotype predictions. For some antibiotics, such as ciprofloxacin, resistance is predictable by target site mutations in a single gene, *gyrA*[3,5]. However, recent efforts to predict azithromycin minimum inhibitory concentrations (MICs) using regression-based or machine-learning approaches have indicated that a substantial fraction of phenotypic resistance is unexplained, particularly among strains with lower-level resistance[3,7,8]. An improved understanding of the genetic mechanisms and evolutionary pathways to macrolide resistance will therefore be critical for informing the development of diagnostics.

Macrolides function by binding to the 50S ribosome and inhibiting protein synthesis[9]. Increased resistance can occur in *N. gonorrhoeae* through target site modification, primarily via 23S rRNA mutations C2611T[10] and A2059G[11], and through efflux pump upregulation. The main efflux pump associated with antibiotic resistance in the gonococcus is the Mtr efflux pump, comprising a tripartite complex encoded by the *mtrCDE* operon under the regulation of the MtrR repressor and the MtrA activator[1,12–17]. Active site or frameshift mutations in the coding sequence of *mtrR* and promoter mutations in the *mtrR* promoter upregulate *mtrCDE* and result in increased macrolide resistance[1,18]. Mosaic sequences originating from recombination with homologs from commensal *Neisseria* donors can also result in structural changes to *mtrD* and increased expression of *mtrCDE*, which synergistically act to confer resistance[19,20].

Here, we use genome-wide association on a global meta-analysis dataset to identify additional genetic variants that confer increased azithromycin resistance in *N. gonorrhoeae*. We find that conventional single-locus bacterial GWAS approaches lead to confounded results and reduced power, but conducting GWAS conditional on known resistance mutations in 23S rRNA reduces linkage-mediated confounding and increases power to recover mutations associated with lower-level resistance. We experimentally validate one such mutation in the 50S ribosomal protein RplD and identify other rare RplD variants associated with resistance, highlighting the ability of conditional bacterial GWAS to discover causal genes for polygenic microbial phenotypes.

## Results

We previously conducted a linear mixed model GWAS using a global meta-analysis collection of 4852 *N. gonorrhoeae* isolates, collected across 15 studies and spanning 65 countries and 38 years[7]. After conducting GWAS on the 4505 isolates with associated azithromycin MICs, we identified highly significant unitigs (i.e., genetic variants generated from de novo assemblies) mapping to the 23S rRNA, associated with increased resistance, and to the efflux pump gene *mtrC*, associated with increased susceptibility and cervical infections[7]. These results highlighted the potential for GWAS to identify novel modifiers of resistance in *N. gonorrhoeae*. However, the characterized mutations did not fully explain azithromycin heritability and thus pointed towards unknown genetic variants.

**Conditional GWAS identifies a resistance mutation in RplD.** To identify these variants, we re-analyzed the GWAS results focusing on the remaining unitigs, which had lower effect sizes and *p*-values closer to the Bonferroni-corrected *p*-value threshold, calculated using the number of unique patterns, of $3.38 \times 10^{-7}$. Numerous variants were significantly associated with increased MICs, many of which mapped to genes (e.g., *hprA*, WHO_F.1254, and *ydfG*) that had not previously been implicated in macrolide resistance in *Neisseria* (Supplementary Data 1). While these signals could represent novel causal resistance genes, we hypothesized that at least some of these variants could have been spuriously driven to association via genetic linkage with the highly penetrant (A2059G: $\beta$, or effect size on the $\log_2$-transformed MIC scale, = 7.14, 95% CI [6.44, 7.84]; C2611T: $\beta = 3.67$, 95% CI [3.46, 3.88]) and population-stratified 23S rRNA resistance mutations (Supplementary Fig. 1). Supporting this hypothesis, $r^2$—a measure of linkage ranging from 0 to 1—between significant variants and 23S rRNA resistance mutations showed a bimodal distribution with a peak at 0.84 and at 0.04 (Supplementary Fig. 2). The three significant variants that mapped to *hprA*, WHO_F.1254, and *ydfG* had elevated $r^2$ values of 0.16, 0.82, and 0.80 respectively; all three variants demonstrated clear phylogenetic overlap with 23S rRNA mutations (Supplementary Fig. 1). Additionally, we did not observe unitigs associated with previously experimentally validated resistance mutations in the *mtrR* promoter[14] or the *mtrCDE* mosaic alleles[19,20], suggesting decreased power to detect known causal variants with lower effect sizes.

To control for the confounding effect of the 23S rRNA mutations, we conducted a conditional GWAS by incorporating additional covariates in our linear mixed model encoding the number of copies (ranging from 0 to 4) of the resistance-conferring 23S rRNA substitutions C2611T and A2059G. We also conditioned on isolate dataset of origin to address potential spurious hits arising from study-specific sequencing methodologies. After conditioning, the previously significant genes linked to 23S rRNA ($r^2 > 0.80$) decreased below the significance threshold, indicating that they were indeed driven to significance by genetic linkage (Fig. 1 and Supplementary Data 2). The most significant variants after the previously reported *mtrC* indel[7] mapped to the *mtrR* promoter ($\beta$, or effect size, = –0.86, 95% CI [–1.05, –0.68]; *p*-value = $5.44 \times 10^{-20}$), encoding the complement of the *mtrR* promoter 1 bp deletion[21], and to *mtrC* ($\beta = 1.23$, 95% CI [0.93, 1.53]; *p*-value = $9.03 \times 10^{-16}$), in linkage with mosaic *mtr* alleles[19,20]. The increased significance of these known efflux pump resistance mutations suggested improved power to recover causal genes with lower effects. Conditioning on dataset did not substantially affect these results but helped to remove other spurious variants arising due to study-specific biases[22] (Supplementary Fig. 3 and Supplementary Data 3).

A glycine to glutamic acid substitution at site 70 of the 50S ribosomal protein L4 (RplD) was significantly associated with increased azithromycin MICs after conducting the conditional GWAS ($\beta = 0.95$, 95% CI [0.68, 1.23]; *p*-value = $1.08 \times 10^{-11}$) (Fig. 1 and Supplementary Data 2). Structural analysis of the *Thermus thermophilus* 50S ribosome complexed with azithromycin suggests that this amino acid is an important residue in macrolide binding (Supplementary Fig. 4), and RplD substitutions at this binding site modulate macrolide resistance in other bacteria[23,24]. This substitution has previously been observed rarely in gonococcus and the association with azithromycin resistance versus susceptibility was non-significant[3,25,26]; as a result, the role of RplD mutations in conferring macrolide resistance was unclear. To assess the contribution of RplD mutations to continuous azithromycin MIC levels, we modeled MICs using a linear regression framework with known genetic

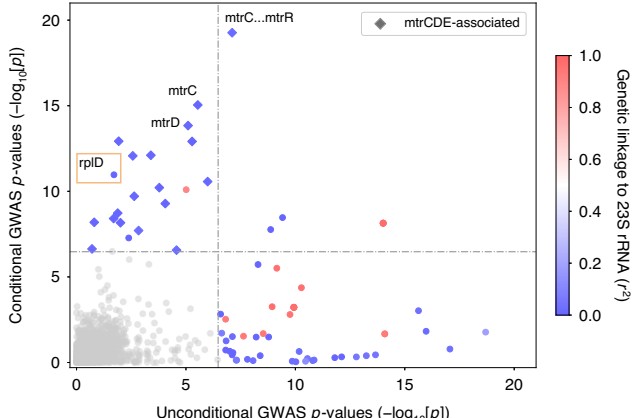

**Fig. 1 GWAS conditional on 23S rRNA mutations and dataset demonstrates decreased confounding and increased power.** Each variant is plotted using negative $\log_{10}$-transformed $p$-values, calculated using likelihood-ratio tests, for the association with azithromycin MICs in unconditional and conditional GWASes. Genetic linkage measured by $r^2$ to 23S rRNA mutations A2059G and C2611T is colored for significant variants as indicated on the right, ranging from 0 (blue) to 0.5 (white) to 1 (red). Variants associated with previously experimentally verified resistance mechanisms in the *mtrR* and *mtrCDE* promoters and coding regions are denoted using diamonds. Bonferroni thresholds, calculated using the number of unique patterns, for both GWASes are depicted using a gray dashed line at $3.38 \times 10^{-7}$. Plot axes are limited to highlight variants associated with lower-level resistance; as a result, the highly significant 23S rRNA substitutions and *mtrC* indel mutations[7] are not shown.

resistance determinants as predictors (Supplementary Data 4 and 5)[7,27]. Compared to this baseline model, inclusion of the RplD G70D mutation decreased the number of strains with unexplained MIC variation (defined as absolute model error greater than one MIC dilution) from 1514 to 1463, improved adjusted $R^2$ from 0.691 to 0.704, and significantly improved model fit ($p$-value $< 10^{-10}$; $\chi^2$ test statistic $= 288.51$; Likelihood-ratio $\chi^2$ test for nested models). These results indicate that RplD G70D is a strong candidate for addressing a portion of the unexplained azithromycin resistance in *N. gonorrhoeae*.

**Genomic epidemiology of RplD macrolide binding site mutations.** We next assessed the population-wide prevalence and diversity of RplD-azithromycin binding site mutations. The RplD G70D mutation was present in 231 out of 4850 isolates (4.76%) with multiple introductions observed across varied genetic backgrounds (Fig. 2). An additional 34 isolates contained mutations at amino acids 68 (G68D, G68C), 69 (T69I), and 70 (G70S, G70A, G70R, G70duplication) (Fig. 3a). These other putative RplD binding site mutations were associated with significantly higher azithromycin MICs compared to both RplD G70D and RplD wild-type strains, indicating multiple avenues for disruption of macrolide binding (Fig. 3b). Grouping all RplD binding site mutations together resulted in increased effect size ($\beta = 1.02$) and $p$-value ($9.25 \times 10^{-18}$) in the conditional GWAS linear mixed model compared to the association with just RplD G70D ($\beta = 0.95$, $p$-value $= 1.08 \times 10^{-11}$). Strains with RplD binding site mutations were identified from 21 countries from 1993 to 2015 with prevalence reaching over 10% in some datasets (New York City 2011–2015[28] and Japan 1996–2015[29]; Supplementary Table 1), in line with sustained transmission of RplD G70D strains (Fig. 2). Our results suggest that macrolide binding to the 50S ribosome can be disrupted via multiple mutations and that

these mutations are widespread contributors to azithromycin resistance in some populations.

**Experimental validation and growth dynamics of RplD G70D strains.** To experimentally verify that RplD G70D contributes to macrolide resistance, we constructed an isogenic derivative of the laboratory strain 28Bl with the G70D substitution (using two biological replicates: C5 and E9) and tested for MIC differences across a panel of macrolides. Azithromycin and erythromycin MICs increased by three-fold, and clarithromycin MICs increased by six-fold on average in the G70D strains compared to the wild-type strain (Table 1). We also compared the experimental results with our modeling analysis: the estimate from our linear model for the azithromycin MIC of a strain that contains the RplD G70D mutation and no other resistance mutations was 0.363, which agrees well with the experimental results. Macrolide resistance has been associated with a fitness cost in other species[30], prompting us to measure the in vitro growth dynamics of the RplD G70D strain. Time-course growth curves of the wild-type strain 28Bl and isogenic G70D strain E9 were similar (Supplementary Fig. 5) with overlapping estimates of doubling times: 28Bl doubling time $= 1.756$ h, 95% CI [1.663, 1.861] versus 28Bl RplD$^{G70D}$ (E9) doubling time $= 1.787$ h, 95% CI [1.671, 1.920] (Supplementary Table 2). These results confirm the role of RplD G70D in mediating macrolide resistance and indicate a lack of severe associated in vitro fitness costs.

**Discussion**
Azithromycin resistance in *N. gonorrhoeae* is a polygenic trait involving contributions from mutations in different 50S ribosomal components, up- and down-regulation of efflux pump activity, and additional unknown factors (Supplementary Table 3). Genome-wide association methods offer one approach for uncovering the genotypic basis of unexplained resistance in clinical isolates, but novel causal genes associated with lower effects have been difficult to identify with traditional microbial GWAS approaches[24]. Our results indicate that extending the GWAS linear mixed model to incorporate known causal genetic variants could address some of these challenges, particularly when known genes exhibit strong penetrance and population stratification, obfuscating signals with lower effects. After conducting conditional GWAS on azithromycin MICs, we observed a reduction in spurious results attributable to genetic linkage with known high-level resistance mutations in the 23S rRNA, and an increase in power to recover secondary resistance mutations in the MtrCDE efflux pump. We also identified a resistance-associated mutation in the macrolide binding site of 50S ribosomal protein RplD as significant only after conditioning. These results are in line with studies of multi-locus methods in the human GWAS field showing increased power[31–33] and complementary methods using whole-genome elastic nets for microbial genome data[34,35].

The situations under which conditional GWAS improves the power to detect new causal genes will need to be further characterized in other bacterial species and through simulations[35]. Here, we observed both increased magnitude of effect (*rplD* $\beta$ increased from 0.52 to 0.95) and increased model precision (*rplD* standard error decreased from 0.223 to 0.140) after conditioning, both of which could improve power. The success of this conditional analysis using a relatively small sample size compared to human GWAS studies may also be attributable to the degree of homoplasy, as RplD mutations have been acquired multiple times across the phylogeny (Fig. 2). The extent of genetic linkage between true positives and the dominant resistance gene is likely key: if the degree of linkage is high (e.g., because a few bacterial

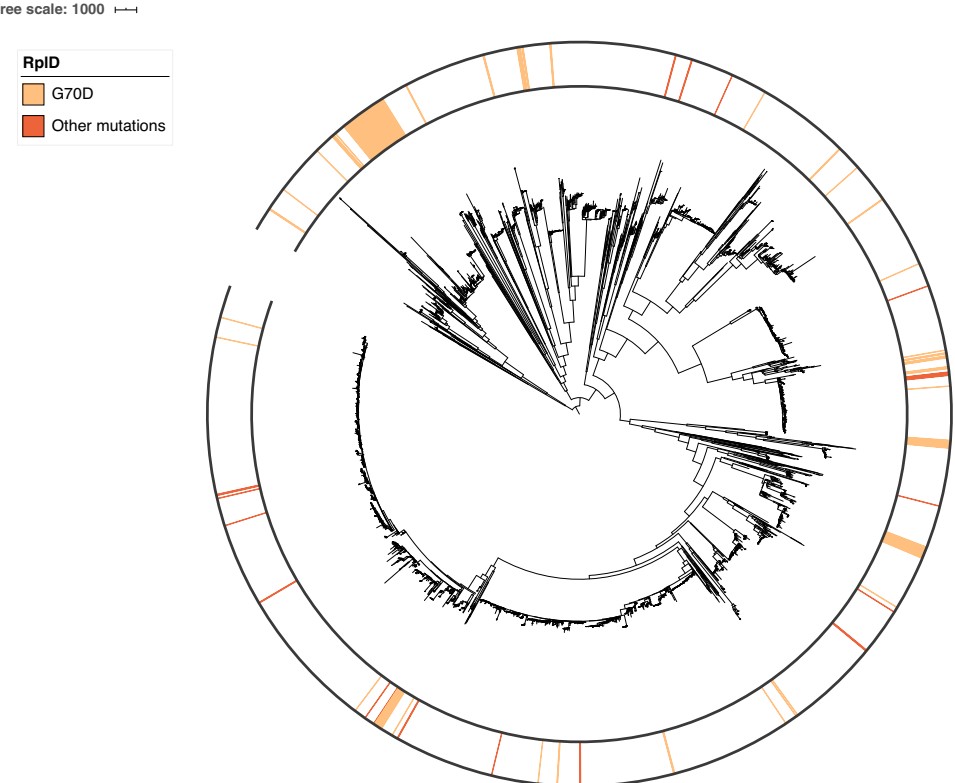

**Fig. 2 Population structure of RplD binding site mutations in a global gonococcal meta-analysis dataset.** A midpoint rooted recombination-corrected maximum likelihood phylogeny of 4852 genomes based on 68697 SNPs non-recombinant from Ma and Mortimer et al.[7] was annotated with the presence of RplD macrolide binding site mutations (orange for G70D and dark orange for other binding site mutations). Branch length represents total number of substitutions after removal of predicted recombination.

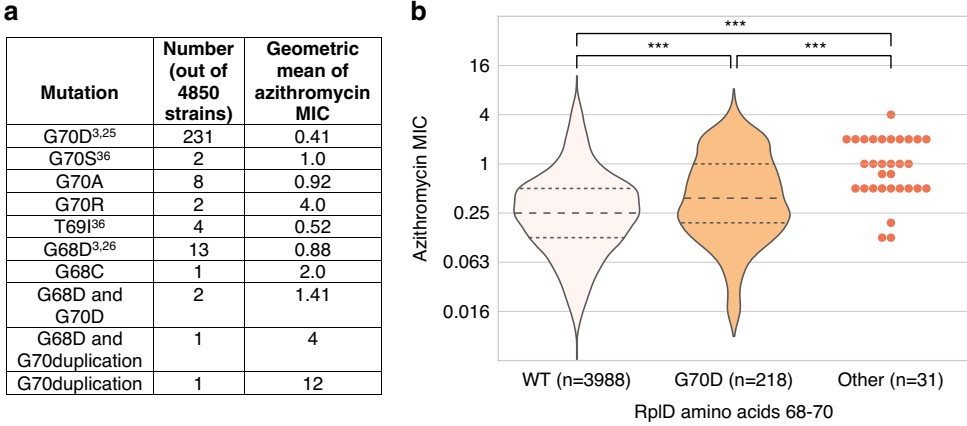

**a**

| Mutation | Number (out of 4850 strains) | Geometric mean of azithromycin MIC |
|---|---|---|
| G70D[3,25] | 231 | 0.41 |
| G70S[36] | 2 | 1.0 |
| G70A | 8 | 0.92 |
| G70R | 2 | 4.0 |
| T69I[36] | 4 | 0.52 |
| G68D[3,26] | 13 | 0.88 |
| G68C | 1 | 2.0 |
| G68D and G70D | 2 | 1.41 |
| G68D and G70duplication | 1 | 4 |
| G70duplication | 1 | 12 |

**Fig. 3 Varied RplD macrolide binding site mutations are associated with increased azithromycin MICs.** Mean (**a**) and distribution (**b**) of azithromycin MICs for RplD macrolide binding site variants. Previously reported mutations are cited with the first reporting publications. Violin and swarm plots and statistical analyses were limited to isolates with MICs < 8 to exclude isolates with 23S rRNA mutations. Rare RplD mutations (denoted as "Other (n = 31)") were grouped for visualization and statistical analysis, and thus were assumed to all have the same effect. Quartiles within violin plots are depicted using dotted lines. Statistical significance between RplD variants and RplD wildtype MIC distributions was assessed by two-sided Mann–Whitney U Test: $*p < 0.05$, $**p < 0.01$, and $***p < 0.001$. Exact p-values from left to right were $3.09 \times 10^{-7}$ (WT vs. G70D), $3.91 \times 10^{-10}$ (WT vs. Other), $5.74 \times 10^{-5}$ (G70D vs. Other).

lineages repeatedly acquire different mechanisms of resistance to the same drug), then a conditional GWAS could lead to a loss of power to detect the causal genes.

The role of RplD G70D mutations in conferring azithromycin resistance has previously been unclear, in part because of its lower effect size relative to 23S rRNA mutations. The G70D mutation was first observed in isolates from France 2013–2014[25] and in the US Centers for Disease Control Gonococcal Isolate Surveillance Program (CDC GISP) surveillance isolates from 2000 to 2013[3], and a related G68D mutation was described in the GISP collection and in European isolates from 2009 to 2014[26]. However, these analyses reported no clear association with categorical

**Table 1 Macrolide MICs of laboratory strain 28Bl and two isogenic derivatives confirms increased macrolide resistance conferred by RplD G70D.**

| Isolate | Azithromycin MIC (µg/mL) | Clarithromycin MIC (µg/mL) | Erythromycin MIC (µg/mL) |
|---|---|---|---|
| 28Bl | 0.094 | 0.25 | 0.38 |
| 28Bl RplD$^{G70D}$ (C5) | 0.25 (2.66×) | 1.5 (6×) | 1.5 (3.94×) |
| 28Bl RplD$^{G70D}$ (E9) | 0.38 (4.04×) | 1.5 (6×) | 1.0 (2.63×) |

Fold change relative to baseline is shown in parentheses. MICs were measured once for each isogenic derivative using Etest strips placed onto GCB agar plates supplemented with 1% IsoVitaleX.

resistance versus susceptibility. In line with this, we observed lower significance for the RplD unitig in the conditional GWAS model when the isolates were dichotomized into azithromycin susceptible versus non-susceptible ($p$-value $= 3.38 \times 10^{-09}$ versus $1.08 \times 10^{-11}$ in the continuous case). Follow up studies in the US, Eastern China, and a historical Danish collection also reported strains with the G70D mutation[36–38], but other surveillance datasets from Canada, Switzerland, and Nanjing did not[10,39–41], indicating geography-specific circulation. As a result of this ambiguity, previous studies modeling phenotypic azithromycin resistance from genotype did not include RplD mutations[27,42].

Here, we provided confirmatory evidence that the RplD G70D mutation increases macrolide MICs several-fold and that inclusion of the mutation in resistance regression models improves model fit, in line with the GWAS analyses. While RplD G70D mutations on their own are not predicted to confer resistance levels above the clinical CLSI non-susceptibility threshold of 1.0 µg/mL, there is growing appreciation of the role that sub-breakpoint increases in resistance can play in mediating treatment failure[43]. For example, treatment failures in Japan after a 2 g azithromycin dose were associated with MICs as low as 0.5 µg/mL[44], and treatment failures in several case studies of patients treated with a 1 g azithromycin dose were associated with MICs of 0.125 to 0.25 µg/mL[45]. Low level azithromycin resistance may also serve as a stepping stone to higher level resistance, as suggested by an analysis of an outbreak of a high level azithromycin resistant *N. gonorrhoeae* lineage in the UK[46].

We also observed multiple previously undescribed mutations in the RplD macrolide binding site associated with even higher MICs than the G70D mutation. The transmission of these isolates has been relatively limited, potentially due to increased fitness costs commensurate with increased resistance. In contrast, several lines of evidence suggest that the G70D mutation carries a relatively minimal fitness cost. Time-course growth experiments indicated that the RplD G70D isogenic pair of strains have similar doubling times, and phylogenetic analyses suggest multiple acquisitions of G70D in distinct genetic backgrounds, with a lineage in New York City showing evidence of sustained transmission.

As macrolide use continues to select for increased resistance in *N. gonorrhoeae*, both the RplD G70D and rarer binding site mutations should be targets for surveillance in future whole-genome sequencing studies. The rapid increase in the prevalence of strains with mosaic *mtr* alleles conferring azithromycin reduced susceptibility underscores how quickly the molecular landscape of resistance can change[47,48] and highlights the value of early and proactive surveillance studies. Systematic genomic surveillance in turn allows for novel resistance mutations to be identified using conditional GWAS and other complementary approaches. With an increasingly refined understanding of the

molecular basis of resistance, sequence-based diagnostics can then be developed by leveraging emerging point-of-care technologies such as Nanopore sequencing and CRISPR-based paper diagnostics[49,50]. The methods here can also be easily extended for other antibiotics used to treat gonococcal infections such as ceftriaxone, where resistance is of paramount concern and molecular mechanisms underlying resistance are still being uncovered[51].

In summary, by reducing genetic confounders and amplifying true signals through bacterial GWAS conditional on known effects, we identified and experimentally characterized mutations in the 50S ribosome that contribute to increased macrolide resistance in *N. gonorrhoeae*.

## Methods

**Genomics and GWAS**. All isolates included in this study are listed in Supplementary Data 6. We conducted whole-genome sequencing assembly, resistance allele calling, phylogenetic inference, genome-wide association, and significant unitig mapping using methods from a prior GWAS[7]. Briefly, reads were downloaded using fastq-dump in SRA toolkit (version 2.8.1). We then created a recombination-corrected phylogeny by running Gubbins (version 2.3.4)[52] on an alignment of pseudogenomes generated from filtered SNPs from Pilon (version 1.16)[53] after mapping reads in BWA-MEM (version 0.7.17-r1188)[54] to the NCCP11945 reference genome (RefSeq accession: NC_011035.1). 23S rRNA mutations were called by mapping reads to a copy of the 23S rRNA locus and analyzing the frequency of variants[55]. Read mapping quality control was conducted in FastQC (version 0.11.7, https://www.bioinformatics.babraham.ac.uk/projects/fastqc/) and BamQC in Qualimap (version 2.2.1)[56], and read deduplication was conducted using Picard (version 2.8.0, https://github.com/broadinstitute/picard). We also annotated assemblies with Prokka (version1.13)[57] and clustered core genes using Roary (version 3.12)[58].

All phylogenies and annotation rings were visualized in iTOL (version 5.5)[59]. As in the prior study, azithromycin MICs prior to 2005 from the CDC GISP dataset[3] were doubled to account for an MIC protocol testing change[60]. For analyses using susceptible versus non-susceptible categories as the outcome variable, isolates with adjusted azithromycin MICs of 1.0 µg/mL or lower were classified as susceptible.

We use a linear mixed model-based GWAS to control for population structure:

$$\mathbf{Y} \sim \mathbf{W}\alpha + \mathbf{X}\beta + \mathbf{u} + \boldsymbol{\epsilon} \tag{1}$$

$$\mathbf{u} \sim N\left(0, \sigma_g^2 \mathbf{K}\right) \tag{2}$$

$$\boldsymbol{\epsilon} \sim N\left(0, \sigma_e^2 \mathbf{I}\right) \tag{3}$$

Here, $\mathbf{Y}$ is the vector of azithromycin MICs, $\mathbf{W}$ is the covariate matrix and $\alpha$ their fixed effects, $\mathbf{X}$ is the genetic variant/unitig under consideration and $\beta$ its fixed effect, $\mathbf{u}$ is a random effect parameterized with population structure matrix $\mathbf{K}$ and additive genetic variance $\sigma_g^2$, and $\boldsymbol{\epsilon}$ is a random effect that models the non-genetic effects parameterized with variance $\sigma_e^2$ and identity matrix $\mathbf{I}$. This model is fit individually for all variants and the $p$-value for $\beta$ is estimated using the likelihood-ratio test. The covariates can include isolate metadata such as country of origin or dataset as well as genetic information encoding known resistance genes.

To conduct the GWAS in Pyseer (version 1.2.0)[61], unitigs were generated from genomes assembled with SPAdes (version 3.12.0)[62] using GATB, and a population structure matrix was generated from the Gubbins phylogeny for the linear mixed model. We conducted conditional GWAS in Pyseer (version 1.2.0)[61] by including additional columns in the covariate file encoding the number of 23S rRNA mutations and including flags --covariates and --use-covariates.

We assessed genetic linkage by calculating $r^2$, or the squared correlation coefficient between two variants defined as $r^2 = (p_{ij} - p_i p_j)^2/(p_i (1 - p_i) \, p_j (1 - p_j))$, where $p_i$ is the proportion of strains with variant $i$, $p_j$ is the proportion of strains with variant $j$, and $p_{ij}$ is the proportion of strains with both variants[63,64]. For a given GWAS variant, we calculated $r^2$ between that variant and the significant unitig from the GWAS mapping to 23 S rRNA C2611T. We repeated the calculation for the same variant but with the unitig mapping to 23 S rRNA A2059G, and took the maximum $r^2$ value from the two calculations.

**Azithromycin MIC regression models**. Azithromycin log-transformed MICs were modeled using a panel of resistance markers[7,65] and country of origin in R (version 3.5.1) using the lm function, with and without inclusion of RplD G70D and proximal mutations:

Model 1: Log_AZI ~ Country + MtrR 39 + MtrR 45 + MtrR LOF + MtrC LOF + MtrR promoter + *mtrCDE* BAPS Cluster + 23S rRNA 2059 + 23S rRNA 2611

Model 2: Log_AZI ~ Country + MtrR 39 + MtrR 45 + MtrR LOF + MtrC LOF + MtrR promoter + *mtrCDE* BAPS Cluster + 23S rRNA 2059 + 23S rRNA 2611 + RplD G70D + RplD other 68-70 mutations

Improvement in model fit was assessed using Anova for likelihood-ratio tests for nested models in R (version 3.5.1). BAPS clusters for *mtrCDE* were called as previously described using FastBAPS (version 1.0.0) as a way to flexibly group resistance-conferring mosaic alleles for inclusion in the regression model as a categorical covariate[7,20]. Variance explained by predictors was calculated using the relaimpo R package (version 2.2.3), which assesses the change in model $R^2$ after inclusion of a predictor. Three approaches were used to calculate this change: the "first" metric compares a model without any predictors to a model with just the predictor of interest, the "last" metric compares a model with all predictors except the one of interest to a model with all predictors, and the "lmg" method averages the change in $R^2$ over all possible model subsets.

**Diversity of RplD macrolide binding site mutations.** We ran BLASTn (version 2.6.0)[66] on the de novo assemblies using a query *rplD* sequence from FA1090 (RefSeq accession: NC_002946.2). *rplD* sequences were aligned using MAFFT (version 7.450)[67]. Binding site mutations were identified after in silico translation of nucleotide alignments in Geneious Prime (version 2019.2.1, https://www.geneious.com). Subsequent analyses identifying prevalence, geometric mean azithromycin MIC, and MIC distribution differences were conducted in Python (version 3.6.5) using the Biopython package (version 1.69)[68] and R (version 3.5.1).

**Experimental validation.** We cultured *N. gonorrhoeae* on GCB agar (Difco) plates supplemented with 1% Kellogg's supplements (GCBK) at 37 °C in a 5% $CO_2$ incubator[69]. We conducted antimicrobial susceptibility testing using Etests (bio-Mérieux) placed onto GCB agar plates supplemented with 1% IsoVitaleX (Becton Dickinson). We selected laboratory strain 28Bl for construction of isogenic strains and measured its MIC for azithromycin, clarithromycin, and erythromycin[20]. *rplD* encoding the G70D mutation was PCR amplified from RplD G70D isolate GCGS1043[3] using primers rplD_FWD_DUS (5′ CATGCCGTCTGAACAA-GACCCGGGTCGCG 3′) (containing a DUS tag to enhance transformation[70]) and rplD_REV (5′ TTCAGAAACGACAGGCGCC 3′). The resulting ~1 kb amplicon was spot transformed[69] into 28Bl. We selected for transformants by plating onto GCBK plates with clarithromycin 0.4 μg/mL and erythromycin 0.4 μg/mL. We confirmed via Sanger sequencing that transformants had acquired the RplD G70D mutation and selected one transformant from each selection condition (strain C5 for clarithromycin and strain E9 for erythromycin) for further characterization. We confirmed that for all macrolides used for selection, no spontaneous resistant mutants were observed after conducting control transformations in the absence of GCGS1043 PCR product. We did not construct strains with the mutation complemented.

**Growth assays.** We streaked 28Bl and 28Bl RplD$^{G70D}$ (E9) onto GCBK plates and grew them overnight for 16 hours at 37 °C in a 5% $CO_2$ atmosphere. We prepared 1 L of fresh Graver Wade (GW) media[71] and re-suspended overnight cultures into 1 mL of GW. After normalizing cultures to OD 0.1, we diluted cultures 1:10$^5$ and inoculated central wells of a 24-well plate with 1.5 mL GW and cells in triplicate. Edge wells were filled with 1.5 mL water. After growth for 1 hour to acclimate to media conditions, we sampled CFUs every 2 hours for a total of 12 hours. For each timepoint, we aspirated using a P1000 micropipette to dissolve clumps and then plated serial dilutions onto a GCBK plate. We counted CFUs the following day and used GraphPad Prism (version 8.2.0 for Windows, GraphPad Software) to graph the data and estimate exponential phase growth rates following removal of lag phase data points and log-transformation of CFUs/mL.

**Reporting summary.** Further information on research design is available in the Nature Research Reporting Summary linked to this article.

## Data availability

In Supplementary Data 6, we have included accession numbers (via publicly hosted database NCBI SRA) for accessing all raw sequence data used for *N. gonorrhoeae* analyses. Intermediate outputs from the genomics pipeline (e.g., de novo assemblies) may also be available from the authors upon request. An interactive and downloadable version of the phylogeny and annotation rings used in Fig. 2 and Supplementary Fig. 1 is hosted at https://itol.embl.de/tree/128103224535135159733824. Source data are available at https://github.com/gradlab/rplD-conditional-GWAS (https://doi.org/10.5281/zenodo.4042334)[72].

## Code availability

Code to reproduce the analyses and figures is available at https://github.com/gradlab/rplD-conditional-GWAS (https://doi.org/10.5281/zenodo.4042334)[72] or from the authors upon request.

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

## Acknowledgements

This work was supported by the NIH/NIAID grant 1R01AI132606-01 and the Smith Family Foundation. T.D.M. is additionally supported by the NIH/NIAID 1 F32 AI145157-01, and K.C.M. is additionally supported by the NSF GRFP grant DGE1745303. Portions of this research were conducted on the O2 high-performance computing cluster, supported by the Research Computing Group at Harvard Medical School. We additionally thank Daniel Rubin, Samantha Palace, Crista Wadsworth, and other members of the Grad Lab for helpful comments during development of the project.

## Author contributions

K.C.M., T.D.M., A.L.H., N.E.W., and L.S.B. performed and interpreted genomic analyses. K.C.M. and M.A.D. performed and interpreted experimental analyses. Y.H.G. supervised the project. K.C.M. and Y.H.G. wrote the paper with contributions from all authors.

## Competing interests

The authors declare no competing interests.

**Additional information**

