## [Peer Review File · Nature Communications]

REVIEWER COMMENTS

Reviewer #1 (Remarks to the Author):

This is an interesting and competently executed study which advances both bacterial GWAS techniques, and our understanding of antibiotic resistance in an important pathogen. The authors apply a technique which has been used to fine-map associations in large human GWAS studies to their curated collection of *N. gonorrhoeae* genomes. The authors show that by conditioning on known associations with macrolide resistance, weaker effects can become significant, and using this approach they discover variants in a ribosomal protein to be associated with resistance. These variants are relatively rare, and yet influential. The authors confirm the association by constructing isogenic mutants with the G70D mutation in two genetic backgrounds.

As the authors note, their discovery will feed into both future predictive models of resistance, and understanding resistance mechanisms in this pathogen. I also believe the methods advanced here will be used in other GWAS studies of antimicrobial resistance.

Comments on scientific content:

- * I think that more explanation of the previous GWAS study (ref 7) is needed at the start of the results section. While the advances of this study are well described, I felt that more background was needed on previous GWAS efforts for this phenotype before diving into novel findings derived by extending previous methods/datasets. More description of the makeup of the dataset itself would also be useful.
- * In a sense, this is a (successful) fine-mapping study with a relatively small sample size compared to what is required for similar efforts in human GWAS, which I think should be highlighted. It seems likely that homoplasmy has increased fine-mapping power and made this causal analysis possible - a key finding.
- * Figure 1: I found it somewhat surprising that the top-right box is empty - why do the *mtr/rplD* p-values only reach significance (in some cases with drastic changes in p-value) after conditioning, despite their low r^2 values? I would (naively) have thought that these p-values would be significant (or change only slightly) before and after conditioning, but that they would be more 'visible' due to the lower number of significant associations. Perhaps this is in a sense a feature of bacterial genomes where $r^2 > 0$ across the genome? Could the authors comment further? To this end, I would also consider adding fig s3 as a panel to this figure - I found it very helpful when considering the conditional GWAS aspect, even though it is not a main result that is taken forwards in the remainder of the paper.
- * l183: 'The estimate from our linear model for the azithromycin MIC of a strain that contains the *RplD* G70D mutation and no other resistance mutations was 0.252, which is in line with the experimental results.' This is a nice comparison to make. Perhaps the authors could add a column to table 1 which shows MIC prediction from the linear for each row, which would be easier to compare.
- * The discussion section would be a good place to summarise the proportion of phenotype variance that has now been explained by each gonococcal genetic mechanism this group have discovered for macrolide resistance and how much missing heritability remains in this dataset. A table listing genes/variants, minor allele frequency, and variance explained would be one option.
- * The authors use a combination of existing packages and their own code to complete the analysis, however this code is not included as part of the manuscript. As much for this code as is feasible should be included with this paper (on github or similar). Particularly, additional code used for the 'genomics and GWAS section' and the R models within should be submitted.
- * Describing precisely how to perform a conditional analysis would be very useful for other authors, and increase the reach of this work. This could be done either by: adding a description to the methods, publishing code to do this, or writing a vignette in the GWAS package documentation itself.

Suggestions (left to the authors' discretion):

* In the section on isogenic mutants, I did not see whether the authors also constructed the appropriate corrected mutant. Also, was table 1 a single measurement of resistance? Even if we expect this phenotype to have little variation in these measurements, this variation should still be quantified and significance calculated.

These are minor points, and I am not suggesting that the authors complete additional lab work, which I do not think is required for publication. But noting this in the methods, or including this information if the authors already have it would be useful.

* What is the significance and effect size if all rplD distribution are grouped as a single test, then input to the conditioned GWAS model? Also, could the authors explain in the text (around l147) what is being assumed by grouping these rare variants together in figure 3, i.e. that they all increase resistance.

* The authors note that previous studies may have missed these results due to only using a resistant/sensitive definition (which should maybe replace the word 'binarized', unless this is a commonly used word in the resistance literature). Can the authors quantify this by repeating their conditioned GWAS at a typical MIC threshold, and compare the p-value with the one arrived at with the current continuous definition.

Minor points:

* Abstract - quoting beta as the main measure of significance isn't those most intuitive, as it is not an odds-ratio, and the scale it is measured on is not defined. p-value and/or MIC increase (with units) would be more useful.

* 'univariate' may be a non standard use, as I think this typically refers to a single response variable, rather than single predictor variable.

* Fig 1: As the authors point out, r^2 is multimodal. I therefore think a diverging colour scale would show these r^2 differences a little more clearly (and would be more similar to those seen in human GWAS, usually produced with locuszoom).

* Fig 2: Consider adding this data and plot to an interactive service such as microreact, to allow users to investigate this large tree in more detail.

* Fig 3: This is an excellent figure which summarises a lot of information clearly. For the G70D and Other groups, it would be convenient to assess the individual points (particularly for $N = 31$). Could the authors try a swarm plot, and maybe add as a supplementary figure if it appears to be helpful?

* Replace 'NYC' with 'New York City' (used twice and not defined)

* The reference to Mortimer et al 2020 appears to be misformatted, and it doesn't have a reference number.

* 'Bonferonni-corrected p-value threshold of 2.97×10^{-7} '. Please explain how this was calculated. Number of tests? Number of unique patterns?

* 'p-value $< 2.2 \times 10^{-16}$; Likelihood-ratio χ^2 test for nested models'. Please give the LRT test statistic, and truncate the precision of this p-value (which in this case has exceeded the precision of a float, something like $p < 10^{-10}$ would probably be more appropriate).

* Methods l274: Add a mathematical definition of the model used with/without conditioning, to demonstrate which terms are different, and whether they are fixed or random effects.

* Methods l285: Explain the rationale and effect of including all interaction terms in these models (and make clear that the \wedge^2 is R syntax to include interactions)

* Methods l290: Explain what 'BAPS groups' are for MtrCDE, and why they have been used.

John Lees

Reviewer #2 (Remarks to the Author):

The article by Ma et al is lovely research that clarifies the role of the G70D and other mutations in the 50S ribosomal protein L4 regarding azithromycin resistance in *N. gonorrhoeae*. The G70D mutation has been previously described, but its role was unclear due to the effects of mutations in 23S rRNA. The authors have performed other elegant studies using genome wide association (GWAS) methods to identify novel mutations associated with antibiotic resistance in *N. gonorrhoeae*. However, these methods have difficulty in identifying mutations with "lower effects" i.e. producing MICs below the breakpoints but which, none-the-less, contribute to increasing MIC values. To remedy this problem, the authors used a condition GWAS method taking into account known resistance mutations (such as 23S rRNA). Interestingly, the G70D mutation did not contribute a fitness cost to the organism as ascertained by growth curve analysis.

The study is methodologically sound with a variety of methods used to confirm results –including transformation studies to prove the effects of the G70 mutation on macrolide resistance using isogenic *N. gonorrhoeae* strains. A well documented data set, previously used by the authors, of 4535 gonococcal genomes, was used for this study. The paper is clearly and succinctly written. This research will impact those studying the incremental increases in MIC to azithromycin in *N. gonorrhoeae* isolates identified in various surveillance studies globally. It contributes to the overall understanding of the complexity of antimicrobial resistance in *N. gonorrhoeae*.

Some minor comments.

Since the majority of the bioinformatics analysis was performed using open source tools, the authors could consider including a supplementary method with line by line commands (and codes under an open license) used for the analysis. This would make their work more transparent and potentially reproducible.

The issue of MIC creep is a longstanding one in the *Neisseria* field. Often, these creeping MICs, in isolates which would otherwise be classified as susceptible, are ignored. Could some of the more important mutations associated with MIC creep be used as an alert system in surveillance to emerging resistance? What other antibiotics resistances should be evaluated using these methods? The authors should add a small discussion regarding the longer term implications of their research. How do the authors envision RplD G70D and other mutations being targets for surveillance? What methods?

Reviewer #3 (Remarks to the Author):

In this manuscript, the authors address an interesting question that is being considered in several pathogenic bacteria. Essentially, comparative genomics approaches have identified genetic determinants of antimicrobial resistance and modern genome-wide association studies (GWAS) have been used to rapidly screen large numbers of bacterial genomes to derive the putative resistance status of isolates. This is beneficial in molecular epidemiology studies and can influence recommended treatments for infection. Furthermore, such approaches suggest that, while antimicrobial resistance may be principally conferred by a small number of genomic changes, in other cases resistance is not fully explained and may be associated with other poorly characterized genomic changes.

This study focusses on *Neisseria gonorrhoeae*, a bacterium in which the emergence of azithromycin resistance complicates treatment. The authors use an existing GWAS approach (Pyseer) to identify resistance-associated SNPs in genes other than those that are well known to confer azithromycin resistance. They achieve this by accounting for linkage with these genes and enhancing the power to detect variants with lower effect sizes by conditioning the GWAS on known

resistance mutations (in 23S rRNA and MtrCDE efflux pump genes) - incorporating additional covariates in a linear mixed model and conditioning on isolate origin to address potential spurious hits arising from study specific sequencing methodologies.

They identified previously reported variation in the mtrR promoter region and discovered a resistance associated rplD mutations that were experimentally confirmed and associated with macrolide binding using protein structure prediction.

I consider this to be a generally well-conceived, well-executed and well-articulated study. Some specific comments include:

1. Understanding of the genetic basis of macrolide resistance is improved, but how significant RplD binding site mutations (compared to dominant resistant SNPs)? Also, are they really that prevalent (present in 5.42% of 4850 isolates)?

2. The study is a reanalysis of data in another paper (Ma et al. bioRxiv. 2020. doi: 10.1101/2020.01.07.896696). As this study is so heavily dependent on previous sampling and genome sequencing, I think it is important that a full list of isolates, metadata, and genome assembly statistics etc. are included. Also, links to the archived genomes (SRA) and/or databases.

Reviewer comments

Reviewer #1 (Remarks to the Author):

This is an interesting and competently executed study which advances both bacterial GWAS techniques, and our understanding of antibiotic resistance in an important pathogen. The authors apply a technique which has been used to fine-map associations in large human GWAS studies to their curated collection of *N. gonorrhoeae* genomes. The authors show that by conditioning on known associations with macrolide resistance, weaker effects can become significant, and using this approach they discover variants in a ribosomal protein to be associated with resistance. These variants are relatively rare, and yet influential. The authors confirm the association by constructing isogenic mutants with the G70D mutation in two genetic backgrounds.

As the authors note, their discovery will feed into both future predictive models of resistance, and understanding resistance mechanisms in this pathogen. I also believe the methods advanced here will be used in other GWAS studies of antimicrobial resistance.

We thank the reviewer for their comprehensive and thoughtful summary of the key points from our manuscript, and for the helpful suggestions below.

Comments on scientific content:

* I think that more explanation of the previous GWAS study (ref 7) is needed at the start of the results section. While the advances of this study are well described, I felt that more background was needed on previous GWAS efforts for this phenotype before diving into novel findings derived by extending previous methods/datasets. More description of the makeup of the dataset itself would also be useful.

We thank the reviewer for this point and have added additional background and context for that study.

“We previously conducted a linear mixed model GWAS using a global meta-analysis collection of 4852 *N. gonorrhoeae* isolates, collected across 15 studies and spanning 65 countries and 38 years¹. After conducting GWAS on the 4505 isolates with associated azithromycin MICs, we identified highly significant unities (i.e., genetic variants generated from de novo assemblies) mapping to the 23S rRNA, associated with increased resistance, and to the efflux pump gene *mtrC*, associated with increased susceptibility and cervical infections¹. These results highlighted the potential for GWAS to identify novel modifiers of resistance in *N. gonorrhoeae*. However, the characterized mutations did not fully explain azithromycin heritability and thus pointed towards unknown genetic variants.

To identify these variants, we re-analyzed the GWAS results focusing on the remaining unities, which had lower effect sizes and p-values closer to the Bonferroni-corrected p-value threshold of 3.38×10^{-7} .”

We have also included a more detailed table of all isolates used, their SRA accession numbers, antibiotic MIC metadata, genetic covariates, and assembly / mapping statistics in Supplementary Data 6.

* In a sense, this is a (successful) fine-mapping study with a relatively small sample size compared to what is required for similar efforts in human GWAS, which I think should be highlighted. It seems likely that homoplasmy has increased fine-mapping power and made this causal analysis possible - a key finding.

We agree that this is an important point worth highlighting and have included this in our discussion:

“Moreover, the success of this conditional analysis using a relatively small sample size compared to human GWAS studies may be attributable to the degree of homoplasy, as RplD mutations have been acquired across the phylogeny (Figure 2).”

* Figure 1: I found it somewhat surprising that the top-right box is empty - why do the mtr/rplD p-values only reach significance (in some cases with drastic changes in p-value) after conditioning, despite their low r^2 values? I would (naively) have thought that these p-values would be significant (or change only slightly) before and after conditioning, but that they would be more 'visible' due to the lower number of significant associations. Perhaps this is in a sense a feature of bacterial genomes where $r^2 > 0$ across the genome? Could the authors comment further?

We speculate that the association between mtr/rplD unitigs and increased MICs is weakened in the baseline GWAS because some of the strains without mtr/rplD unitigs have very high MICs due to the 23S rRNA mutations. The random effect in the linear mixed model does not adequately account for these high effect mutations. The conditional GWAS strengthens the association because strains that are both mtr/rplD negative and 23S rRNA positive effectively have lowered MICs, resulting in an increased magnitude of effect (e.g., for rplD $\beta = 0.52$ in the baseline GWAS and $\beta = 0.95$ after conditioning). Additionally, correcting for model mis-specification can reduce the standard error and improve the precision of the model (e.g., for rplD standard error = 0.223 in the baseline GWAS and standard error = 0.140 after conditioning). These two properties could contribute to the observed increase in power.

The degree of genetic linkage likely affects this phenomenon. Causal resistance genes with high linkage to the 23S rRNA mutations in the baseline GWAS will probably experience a drop in significance after conditioning. Thus, the gain in power depends in part on whether resistance tends to be gained repeatedly by certain lineages or whether resistance is acquired throughout the phylogeny (the latter seems true for gonococcal azithromycin resistance). This also suggests there could be potential drawbacks with conducting GWAS conditional on too many genetic covariates, but the nuances of this will need to be explored in further studies. We have added language to the Discussion on this:

“The situations under which conditional GWAS improves the power to detect new causal genes will need to be further characterized in other bacterial species and through simulations². The success of this conditional analysis using a relatively small sample size compared to human GWAS studies may be attributable to the degree of homoplasy, as RplD mutations have been acquired across the phylogeny (Figure 2). The degree of genetic linkage between true positives and the dominant resistance gene is also key: if the degree of linkage is high (e.g., because a few bacterial lineages repeatedly acquire different mechanisms of resistance to the same drug), then a conditional GWAS could lead to a loss of power to detect the causal genes.”

Finally, we think the observation that power can increase from conditioning on large effect size, known loci is not limited to just bacterial genomes as prior work by Segura et al. and Ma et al. among others has shown similar results for human GWAS analyses.

To this end, I would also consider adding fig s3 as a panel to this figure - I found it very helpful when considering the conditional GWAS aspect, even though it is not a main result that is taken forwards in the remainder of the paper.

We experimented with adding Figure S3 as a panel but felt that it would complicate the message around the major advance of the study, which was conducting GWAS conditional on genetic covariates. Our interpretation of Figure S3 is that conditioning additionally on dataset helps to remove a few other probable false positives (and that conditioning on dataset was not what led to the observed power increases and high r^2 false positive eliminations), but the importance of conditioning on isolate country of origin or dataset to remove study-specific spurious results has already been demonstrated nicely previously (e.g. in Lees et al., 2016).

* I183: 'The estimate from our linear model for the azithromycin MIC of a strain that contains the RplD G70D mutation and no other resistance mutations was 0.252, which is in line with the experimental results.' This is a nice comparison to make. Perhaps the authors could add a column to table 1 which shows MIC prediction from the linear for each row, which would be easier to compare.

While we agree that this is a nice result, Table 1 contains multiple experimental results for different macrolide drugs that we did not model in the linear regression model. Thus, we opted to add wording to emphasize this result more in the text, rather than adding a column to this table with just the predicted azithromycin value for an RplD strain.

* The discussion section study would be a good place to summarise the proportion of phenotype variance that has now been explained by each gonococcal genetic mechanism this group have discovered for macrolide resistance and how much missing heritability remains in this dataset. A table listing genes/variants, minor allele frequency, and variance explained would be one option.

We agree this would be helpful for readers and have added a Supplementary Table 3 reporting on the variance explained by the known mechanisms of azithromycin resistance: "Azithromycin resistance in *N. gonorrhoeae* is a polygenic trait involving contributions from mutations in different 50S ribosomal components, up- and down-regulation of efflux pump activity, and additional unknown factors (Supplementary Table 3)."

* The authors use a combination of existing packages and their own code to complete the analysis, however this code is not included as part of the manuscript. As much for this code as is feasible should be included with this paper (on github or similar). Particularly, additional code used for the 'genomics and GWAS section' and the R models within should be submitted.

* Describing precisely how to perform a conditional analysis would be very useful for other authors, and increase the reach of this work. This could be done either by: adding a description to the methods, publishing code to do this, or writing a vignette in the GWAS package documentation itself.

We thank the reviewer for this important suggestion and agree that including the code will improve reproducibility and reach. We have created a Github repository and notebook file that steps through all analyses and figures in the paper, including further description of how to conduct a conditional GWAS analysis, and have included data and code availability statements in the manuscript that link there:

Data availability

In Supplementary Data 6, we have included accession numbers (via publicly hosted database NCBI SRA) for accessing all raw sequence data used for *N. gonorrhoeae* analyses. Intermediate outputs from the genomics pipeline (e.g., de novo assemblies) may also be available from the authors upon request. An interactive and downloadable version of the phylogeny and annotation rings used in Figure 2 and Supplementary Figure 1 is hosted at <https://itol.embl.de/tree/1281032245351351597338246>. Source data underlying all figures are available in the Supplementary Data or at <https://github.com/gradlab/rpID-conditional-GWAS>.

Code availability

Code to reproduce the analyses and figures is available at <https://github.com/gradlab/rpID-conditional-GWAS> or from the authors upon request.”

Suggestions (left to the authors' discretion):

* In the section on isogenic mutants, I did not see whether the authors also constructed the appropriate corrected mutant. Also, was table 1 a single measurement of resistance? Even if we expect this phenotype to have little variation in these measurements, this variation should still be quantified and significance calculated.

These are minor points, and I am not suggesting that the authors complete additional lab work, which I do not think is required for publication. But noting this in the methods, or including this information if the authors already have it would be useful.

We have updated the caption of Table 1 to specify that Etests “were measured once for each isogenic derivative” and the methods to indicate that we “did not construct strains with the mutation complemented”.

* What is the significance and effect size if all rpID distribution are grouped as a single test, then input to the conditioned GWAS model? Also, could the authors explain in the text (around l147) what is being assumed by grouping these rare variants together in figure 3, i.e. that they all increase resistance.

We tested the effect of conducting a burden test on RpID and have included this in the results: “Grouping all RpID binding site mutations together resulted in increased effect size ($\beta = 1.02$) and p-value (9.25×10^{-18}) in the conditional GWAS linear mixed model compared to the association with just RpID G70D ($\beta = 0.95$, p-value = 1.08×10^{-11}).”

We have added the following sentence to the caption of Figure 3: “Rare RpID mutations (denoted as “Other (n=31)”) were grouped for visualization and statistical analysis, and thus were assumed to all have the same effect.”

* The authors note that previous studies may have missed these results due to only using a resistant/sensitive definition (which should maybe replace the word 'binarized', unless this is a commonly used word in the resistance literature). Can the authors quantify this by repeating their

conditioned GWAS at a typical MIC threshold, and compare the p-value with the one arrived at with the current continuous definition.

We have quantified this and included it in the results. We have also adjusted the wording in the section for the resistant/sensitive definition: “However, these analyses reported no clear association with categorical resistance versus susceptibility. In line with this, we observed lower significance for the RplD unitig in the conditional GWAS model when the isolates were dichotomized into azithromycin susceptible versus non-susceptible (p-value = 3.38×10^{-09} versus 1.08×10^{-11} in the continuous case).”

Minor points:

* Abstract - quoting beta as the main measure of significance isn't those most intuitive, as it is not an odds-ratio, and the scale it is measured on is not defined. p-value and/or MIC increase (with units) would be more useful.

We have included the p-value from the conditional GWAS instead of beta.

* 'univariate' may be a non standard use, as I think this typically refers to a single response variable, rather than single predictor variable.

We have replaced the term “univariate” with single variant or single-locus.

* Fig 1: As the authors point out, r^2 is multimodal. I therefore think a diverging colour scale would show these r^2 differences a little more clearly (and would be more similar to those seen in human GWAS, usually produced with locuszoom).

We tried a diverging color scale and agree that the results better highlight the differences in r^2 values. The color schemes have been updated in Figure 1 and Supplementary Figure 2.

* Fig 2: Consider adding this data and plot to an interactive service such as microreact, to allow users to investigate this large tree in more detail.

We have hosted the tree on iTOL and added a data availability statement: “An interactive and downloadable version of the phylogeny and annotation rings used in Figure 1 and Supplementary Figure 1 is hosted at <https://itol.embl.de/tree/1281032245351351597338246>.”

* Fig 3: This is an excellent figure which summarises a lot of information clearly. For the G70D and Other groups, it would be convenient to assess the individual points (particularly for N = 31). Could the authors try a swarm plot, and maybe add as a supplementary figure if it appears to be helpful?

We agree that showing the individual data points for the rarer active site mutations would be helpful and have updated Figure 3 accordingly.

* Replace 'NYC' with 'New York City' (used twice and not defined)

'NYC' has now been replaced with 'New York City'.

* The reference to Mortimer et al 2020 appears to be misformatted, and it doesn't have a reference number.

A correctly formatted reference has now been included.

* 'Bonferonni-corrected p-value threshold of 2.97×10^{-7} '. Please explain how this was calculated. Number of tests? Number of unique patterns?

All mentions of the p-value threshold have been edited to also include methodology (i.e. "calculated using the number of unique patterns").

* 'p-value $< 2.2 \times 10^{-16}$; Likelihood-ratio χ^2 test for nested models'. Please give the LRT test statistic, and truncate the precision of this p-value (which in this case has exceeded the precision of a float, something like $p < 10^{-10}$ would probably be more appropriate).

We have updated the line accordingly: "and significantly improved model fit (p-value $< 10^{-10}$; χ^2 test statistic = 288.51; Likelihood-ratio χ^2 test for nested models)".

* Methods I274: Add a mathematical definition of the model used with/without conditioning, to demonstrate which terms are different, and whether they are fixed or random effects.

The mathematical model has now been specified in the Methods:

"We use a linear mixed model-based GWAS to control for population structure:

$$Y \sim W\alpha + X\beta + u + \epsilon$$

$$u \sim N(0, \sigma_g^2 K)$$

$$\epsilon \sim N(0, \sigma_e^2 I)$$

Here, Y is the vector of azithromycin MICs, W is the covariate matrix and α their fixed effects, X is the genetic variant / unitig under consideration and β its fixed effect, u is a random effect parameterized with population structure matrix K and additive genetic variance σ_g^2 , and ϵ is a random effect that models the non-genetic effects. This model is fit individually for all variants. The covariates can include isolate metadata such as country of origin or dataset as well as genetic information encoding known resistance genes."

* Methods I285: Explain the rationale and effect of including all interaction terms in these models (and make clear that the ^2 is R syntax to include interactions)

* Methods I290: Explain what 'BAPS groups' are for MtrCDE, and why they have been used.

We have updated this section in the Methods with additional detail on the linear regression models and BAPS groups. We also removed the pairwise interaction part of the model because we felt that indiscriminate inclusion of all interaction terms in the regression would not be appropriate.

“Azithromycin log-transformed MICs were modeled using a panel of resistance markers^{1,3} and country of origin in R (version 3.5.1) using the lm function, with and without inclusion of RplD G70D and proximal mutations:

Model 1: $\text{Log_AZI} \sim \text{Country} + \text{MtrR } 39 + \text{MtrR } 45 + \text{MtrR LOF} + \text{MtrC LOF} + \text{MtrR promoter} + \text{mtrCDE BAPS Cluster} + \text{23S rRNA } 2059 + \text{23S rRNA } 2611$

Model 2: $\text{Log_AZI} \sim \text{Country} + \text{MtrR } 39 + \text{MtrR } 45 + \text{MtrR LOF} + \text{MtrC LOF} + \text{MtrR promoter} + \text{mtrCDE BAPS Cluster} + \text{23S rRNA } 2059 + \text{23S rRNA } 2611 + \text{RplD G70D} + \text{RplD other } 68\text{-}70 \text{ mutations}$

Improvement in model fit was assessed using Anova for likelihood-ratio tests for nested models in R (version 3.5.1). BAPS clusters for mtrCDE were called as previously described using FastBAPS (version 1.0.0) as a way to flexibly group resistance-conferring mosaic alleles for inclusion in the regression model as a categorical covariate^{1,4}.”

John Lees

Reviewer #2 (Remarks to the Author):

The article by Ma et al is lovely research that clarifies the role of the G70D and other mutations in the 50S ribosomal protein L4 regarding azithromycin resistance in *N. gonorrhoeae*. The G70D mutation has been previously described, but its role was unclear due to the effects of mutations in 23S rRNA. The authors have performed other elegant studies using genome wide association (GWAS) methods to identify novel mutations associated with antibiotic resistance in *N. gonorrhoeae*. However, these methods have difficulty in identifying mutations with “lower effects” i.e. producing MICs below the breakpoints but which, none-the-less, contribute to increasing MIC values. To remedy this problem, the authors used a condition GWAS method taking into account known resistance mutations (such as 23S rRNA). Interestingly, the G70D mutation did not contribute a fitness cost to the organism as ascertained by growth curve analysis.

The study is methodologically sound with a variety of methods used to confirm results –including transformation studies to prove the effects of the G70 mutation on macrolide resistance using isogenic *N. gonorrhoeae* strains. A well documented data set, previously used by the authors, of 4535 gonococcal genomes, was used for this study. The paper is clearly and succinctly written. This research will impact those studying the incremental increases in MIC to azithromycin in *N. gonorrhoeae* isolates identified in various surveillance studies globally. It contributes to the overall understanding of the complexity of antimicrobial resistance in *N. gonorrhoeae*.

We thank the reviewer for their comprehensive and thoughtful summary of the key points from our manuscript, and for the helpful suggestions below.

Some minor comments.

Since the majority of the bioinformatics analysis was performed using open source tools, the authors could consider including a supplementary method with line by line commands (and codes under an open license) used for the analysis. This would make their work more transparent and potentially reproducible.

We thank the reviewer for this important suggestion and agree that including the code will improve reproducibility and reach. We have created a Github repository and notebook file that steps through all analyses and figures in the paper, including further description of how to conduct a conditional GWAS analysis, and have included data and code availability statements in the manuscript that link there:

“Data availability

In Supplementary Data 6, we have included accession numbers (via publicly hosted database NCBI SRA) for accessing all raw sequence data used for *N. gonorrhoeae* analyses. Intermediate outputs from the genomics pipeline (e.g., de novo assemblies) may also be available from the authors upon request. An interactive and downloadable version of the phylogeny and annotation rings used in Figure 2 and Supplementary Figure 1 is hosted at <https://itol.embl.de/tree/1281032245351351597338246>. Source data underlying all figures are available in the Supplementary Data or at <https://github.com/gradlab/rpID-conditional-GWAS>.

Code availability

Code to reproduce the analyses and figures is available at <https://github.com/gradlab/rpID-conditional-GWAS> or from the authors upon request.”

The issue of MIC creep is a longstanding one in the *Neisseria* field. Often, these creeping MICs, in isolates which would otherwise be classified as susceptible, are ignored. Could some of the more important mutations associated with MIC creep be used as an alert system in surveillance to emerging resistance? What other antibiotics resistances should be evaluated using these methods? The authors should add a small discussion regarding the longer term implications of their research.

How do the authors envision RplD G70D and other mutations being targets for surveillance? What methods?

We share the reviewer’s enthusiasm for these applications and have added further information in the discussion addressing these potential follow up studies:

“As macrolide use continues to select for increased resistance in *N. gonorrhoeae*, both the RplD G70D and rarer binding site mutations should be targets for surveillance in future whole-genome sequencing studies. The rapid increase in the prevalence of strains with mosaic *mtr* alleles conferring azithromycin reduced susceptibility underscores how quickly the molecular landscape of resistance can change^{5,6} and highlights the value of early and proactive surveillance studies. Systematic genomic surveillance in turn allows for novel resistance mutations to be identified using conditional GWAS and other complementary approaches. With an increasingly refined understanding of the molecular basis of resistance, sequence-based diagnostics can then be developed by leveraging emerging point-of-care technologies such as Nanopore sequencing and CRISPR-based paper diagnostics^{7,8}. The methods here can also be easily extended for other antibiotics used to treat gonococcal infections such as ceftriaxone, where resistance is of paramount concern and molecular mechanisms underlying resistance are still being uncovered⁹.”

Reviewer #3 (Remarks to the Author):

In this manuscript, the authors address an interesting question that is being considered in several pathogenic bacteria. Essentially, comparative genomics approaches have identified genetic

determinants of antimicrobial resistance and modern genome-wide association studies (GWAS) have been used to rapidly screen large numbers of bacterial genomes to derive the putative resistance status of isolates. This is beneficial in molecular epidemiology studies and can influence recommended treatments for infection. Furthermore, such approaches suggest that, while antimicrobial resistance may be principally conferred by a small number of genomic changes, in other cases resistance is not fully explained and may be associated with other poorly characterized genomic changes.

This study focusses on *Neisseria gonorrhoeae*, a bacterium in which the emergence of azithromycin resistance complicates treatment. The authors use an existing GWAS approach (Pyseer) to identify resistance-associated SNPs in genes other than those that are well known to confer azithromycin resistance. They achieve this by accounting for linkage with these genes and enhancing the power to detect variants with lower effect sizes by conditioning the GWAS on known resistance mutations (in 23S rRNA and MtrCDE efflux pump genes) - incorporating additional covariates in a linear mixed model and conditioning on isolate origin to address potential spurious hits arising from study specific sequencing methodologies.

They identified previously reported variation in the *mtrR* promoter region and discovered a resistance associated *rplD* mutations that were experimentally confirmed and associated with macrolide binding using protein structure prediction.

We thank the reviewer for their comprehensive and thoughtful summary of the key points from our manuscript, and for the helpful suggestions below.

I consider this to be a generally well-conceived, well-executed and well-articulated study. Some specific comments include:

1. Understanding of the genetic basis of macrolide resistance is improved, but how significant RplD binding site mutations (compared to dominant resistant SNPs)? Also, are they really that prevalent (present in 5.42% of 4850 isolates)?

We thank the reviewer for these important questions. The effect of RplD mutations is less pronounced compared to 23S rRNA mutations but we argue that they are still critical for surveillance given that 1) the correlation between macrolide MIC and treatment failure is still not well understood and evidence exists that failure may occur at sub-breakpoint MICs^{10,11}, 2) higher-level resistance can emerge on the background of lower-level resistance¹², 3) even small increases in performance will help sequence-based diagnostics meet the stringent criteria necessary for approval, and 4) the significance of these mutations can increase as their prevalence also changes.

Prevalence can change or vary in multiple ways. As we've shown in Supplementary Table 1, there is considerable geographic heterogeneity in the prevalence of RplD mutations and in some regions the prevalence is already over 10%. Additionally, prevalence can rapidly increase over time, as suggested by data on mosaic *mtr* alleles that confer azithromycin reduced susceptibility. Isolates with these alleles were first identified in 2012, but surveillance datasets in 2017 from Australia and the US now indicate that these alleles are responsible for 86% and 69% of azithromycin decreased susceptibility, respectively. These data therefore emphasize the importance of early identification and surveillance of panels of resistance mutations before these mutations already become fixed in a population.

As these questions may be shared by readers, we have added language to the Discussion addressing these points.

2. The study is a reanalysis of data in another paper (Ma et al. bioRxiv. 2020. doi: 10.1101/2020.01.07.896696). As this study is so heavily dependent on previous sampling and genome sequencing, I think it is important that a full list of isolates, metadata, and genome assembly statistics etc. are included. Also, links to the archived genomes (SRA) and/or databases."

We agree this would be a helpful resource and have included a table with all isolates used, their SRA accession numbers, azithromycin MICs, all genetic covariates used, and assembly / mapping statistics in Supplementary Data 6.

OTHER CHANGES

During the course of manuscript revision, we noticed a small discrepancy in the number of samples included in the linear models and GWAS (n=4882) and the number of samples included in the other RplD analyses (n=4852). This was due to 30 isolates which should have been left out of the GWAS due to failing to meet genomics quality controls. We re-ran the analysis pipeline with the corrected dataset of n=4852 and confirmed that these did not affect any of the conclusions in the paper. We have updated the manuscript accordingly with the minor adjustments to prevalence estimates, regression estimates, p-values, and figures.

References

- 1 Ma, K. C. et al. Increased antibiotic susceptibility in *Neisseria gonorrhoeae* through adaptation to the cervical environment. *bioRxiv*, doi:10.1101/2020.01.07.896696 (2020).
- 2 Saber, M. & Shapiro, B. Benchmarking bacterial genome-wide association study methods using simulated genomes and phenotypes. *Microbial Genomics*, doi:doi:10.1099/mgen.0.000337 (2020).
- 3 Demczuk, W. et al. Equations To Predict Antimicrobial MICs in *Neisseria gonorrhoeae* Using Molecular Antimicrobial Resistance Determinants. *Antimicrob Agents Chemother* 64, doi:10.1128/AAC.02005-19 (2020).
- 4 Wadsworth, C. B., Arnold, B. J., Sater, M. R. A. & Grad, Y. H. Azithromycin Resistance through Interspecific Acquisition of an Epistasis-Dependent Efflux Pump Component and Transcriptional Regulator in *Neisseria gonorrhoeae*. *MBio* 9, doi:10.1128/mBio.01419-18 (2018).
- 5 Williamson, D. A. et al. Bridging of *Neisseria gonorrhoeae* lineages across sexual networks in the HIV pre-exposure prophylaxis era. *Nat Commun* 10, 3988, doi:10.1038/s41467-019-12053-4 (2019).
- 6 Gernert, K. M. S., Sandra; Schmerer, Matthew W; Thomas IV, Jesse C; Pham, Cau D; St Cyr, Sancta; Schlanger, Karen; Weinstock, Hillard; Shafer, William M; Raphael, Brian H; Kersh, Ellen N; Hun, Sopheay; Hua, Chi; Ruiz, Ryan; Soge, Olusegun O; Dominguez, Catherine; Patel, Ami; Loomis, Jillian; Leavitt, John; Zhang, Jenny; Baldwin, Tamara; Wang, Chun; Moore, Christina; Whelen, Christian; O'Brien, Pamela; Harvey, Alesia. Azithromycin susceptibility of *Neisseria gonorrhoeae* in the USA in 2017: a genomic analysis of surveillance data. *The Lancet Microbe* 1, e154-e164, doi:10.1016/S2666-5247(20)30059-8 (Aug 01, 2020).
- 7 Břinda, K. et al. Rapid heuristic inference of antibiotic resistance and susceptibility by genomic neighbor typing. *bioRxiv*, doi:10.1101/403204 (2019).
- 8 Gootenberg, J. S. et al. Nucleic acid detection with CRISPR-Cas13a/C2c2. *Science* 356, 438-442, doi:10.1126/science.aam9321 (2017).
- 9 Palace, S. G. et al. RNA polymerase mutations cause cephalosporin resistance in clinical *Neisseria gonorrhoeae* isolates. *Elife* 9, doi:10.7554/eLife.51407 (2020).
- 10 Yasuda, M., Ito, S., Hatazaki, K. & Deguchi, T. Remarkable increase of *Neisseria gonorrhoeae* with decreased susceptibility of azithromycin and increase in the failure of azithromycin therapy in male gonococcal urethritis in Sendai in 2015. *J Infect Chemother* 22, 841-843, doi:10.1016/j.jiac.2016.07.012 (2016).
- 11 Tapsall, J. W. et al. Failure of azithromycin therapy in gonorrhea and dis correlation with laboratory test parameters. *Sex. Transm. Dis.* 25, 505-508, doi:10.1097/00007435-199811000-00002 (1998).
- 12 Fifer, H. et al. Sustained transmission of high-level azithromycin-resistant *Neisseria gonorrhoeae* in England: an observational study. *Lancet Infect. Dis.*, doi:10.1016/S1473-3099(18)30122-1 (2018).

REVIEWERS' COMMENTS

Reviewer #1 (Remarks to the Author):

This is a thorough revision, and the authors have addressed all of my previous points and suggestions. I am particularly impressed by the improvement to the methods, and the inclusion of code/notebooks to reproduce all of the analysis.

John Lees